# Local Climate Zones, Sky View Factor and Magnitude of Daytime/Nighttime Urban Heat Islands in Balneário Camboriú, SC, Brazil

**Ismael Luiz Hoppe [1], Cassio Arthur Wollmann [1,\*] , André Schroder Buss [1], João Paulo Assis Gobo [2] and Salman Shooshtarian [3]**

[1] Department of Geosciences, Natural and Exact Sciences Center, Federal University of Santa Maria, Santa Maria 97105-900, Brazil
[2] Department of Geography, Core of Exact Earth Sciences, Federal University of Rondônia, Porto Velho 76801-059, Brazil
[3] School of Property, Construction and Project Management, RMIT University, Melbourne, VIC 3001, Australia
[\*] Correspondence: cassio@ufsm.br

**Abstract:** For this study on urban climatology, the study area is the city of Balneário Camboriú, belonging to the Brazilian state of Santa Catarina (SC), located at 26°59′42″ south latitude and 48°37′46″ west longitude. As it is the most vertical city in the entire Southern Hemisphere, Balneário Camboriú was selected as the study area for the development of this climate analysis. Then, this study was concerned with analyzing the formation of urban heat islands throughout the daytime and nighttime in the city of Balneário Camboriú, Santa Catarina, Brazil, on some days in October 2020, from the perspective of the local climatic zones. Seven fixed sampling points and one official weather station were selected for this research. These points were selected in order to facilitate analysis of the climatic behaviour of the urban area throughout the day, comparing it with the other points, and also to verify possible changes in the local climate in the most diverse types of LCZ. At these same points, the Sky View Factor (SVF) measurements were taken. to elaborate the map of LCZ of Balneário Camboriú, the WUDAPT method was used. There was a great variation of the SVF between the collection points, and different LCZs were mapped, which contributed to the formation of urban heat islands whose maximum magnitude was 10.8 °C and islands with freshnesses of magnitudes of −4.5 °C.

**Keywords:** Local Climate Zones; Sky View Factor; urban heat island; magnitude of UHI; Balneário Camboriú

## 1. Introduction

The world has experienced an increase in urban development over the past two centuries. With the progress of large urban centers, the urban structure has also changed. Thus, urbanization can be defined as the process of temporal, spatial and sectoral changes in the demographic, social, cultural, technological and environmental aspects of a given society [1].

Since the beginning of the nineteenth century, the urban heat islands (UHI) phenomenon has been studied as one of the most prevalent effects of increased urbanization [2]. The UHI is characterized by higher urban air temperatures (Ta) compared to rural temperatures [3]. Large cities experience higher intensities of urban heat islands as a result of urbanization's fast expansion [4].

In the last decade, a study demonstrated that urban–rural techniques are incapable of providing quantitative metadata on site-specific exposure or land cover. The researchers relied primarily on urban and rural qualifiers to characterize the measured landscape [5]. After noticing shortcomings in various studies on urban heat islands, a classification based on local climatic zones (LCZs) and rural areas was created [5].

The LCZ is worldwide, adaptable [6], and simple to comprehend, providing an objective approach for measuring the impact of UHIs in any city. The words "urban" and

"rural" do not adequately and totally characterize the landscape or its environs. Therefore, the review of the definitions of urban and rural landscapes through the LCZ offers two types: city construction (building materials and even verticalization) and land cover type [4–6].

In cities with limited territory, a circumstance which makes them dense and produces a high degree of verticalization, this phenomenon forms cliffs of buildings known as urban canyons, which interferes with the heating of intra-urban air due to solar radiation that is unequally absorbed throughout the day by urban surfaces, such as walls, roofs, and roads [7,8].

The significance of the lowering of the thermal mass maintained in urban street canyons in tropical locations that are shaded during day is shifted by surrounding tall buildings [9]. The sky view factor (SVF), which represents the fraction of the sky viewable from ground level, must be modest. Thus, the canyon base cannot be too deep to give shade and must be shallow enough to prevent radiant heat loss at night. Several research studies on urban heat islands and urban canyons have utilized this methodology [10–13].

In order to advance research on LCZs and UHI [14], especially in southern Brazil, this study analyses the formation of urban heat islands throughout the daytime and nighttime in the city of Balneário Camboriú, Santa Catarina, Brazil, from 14 October 2020 to 22 October 2020, from the perspective of the local climatic zones (LCZ) (Spring in the Southern Hemisphere).

This short collection period was selected because the city was going through a month of successive lockdowns due to the COVID-19 pandemic. Thus, during a small-time window opened by the national health authorities, it was possible to collect the data, installing the meteorological mini shelters. It should be mentioned that this study is still in its early stages on the topic of the urban climate in Balneário Camboriú. Since 2019, the study team has been constructing a much deeper inquiry in this metropolis, which is the most verticalized in the entire Southern Hemisphere, with a wider network of equipment, including meteorological mini shelters and automatic weather stations, Ta, RH, rain, particulate matter floating in the atmosphere, and wind data, and radiation samples were collected with the goal of further research, an unprecedented measurement in this metropolis. This research group maintains a presence on the social network Instagram (@bcc.project), where news on the project's progress, research, and researchers engaged are always updated.

## 2. Materials and Methods

### 2.1. Study Area

The study area is the city of Balneário Camboriú, belonging to the Brazilian state of Santa Catarina (SC), which is located at $-26°59'42''$ latitude and $-48°37'46''$ longitude, in the Mesoregion of Vale do Itajaí and which belongs to the micro-region of Itajaí [15]. Balneário Camboriú is located on the North Coast of Santa Catarina [16]. Balneário Camboriú borders three municipalities, namely Itajaí (north), Itapema (south), and Camboriú (west), and the Atlantic Ocean (east). The main access to the municipality is the BR 101 road, which passes through the west of the municipality and borders the municipalities of Camboriú and Balneário Camboriú (Figure 1).

The population forecast for 2021 is 149,227 according to the Brazilian Institute of Geography and Statistics [15,16]. The entire population resides in the Balneário Camboriú urban area, the area of which is 46.80 km$^2$ with a population density of 2337.67 individuals per square kilometer and a human development index (HDI) of 0.845 [15,16].

Taking into account the hypsometry of the research area, the urban area is 0 to 24 m above sea level. Up to 580 m above sea level, the neighbouring regions of the urban area have a higher altitude. Regarding the Köppen classification for Brazil [17], the Balneário Camboriú climate is classed as Cfa, a humid subtropical climate with hot summers. The average annual temperature ranges from 18.0 °C to 20.0 °C, and the average annual precipitation is around 1768.0 mm, which is evenly spread throughout the seasons.

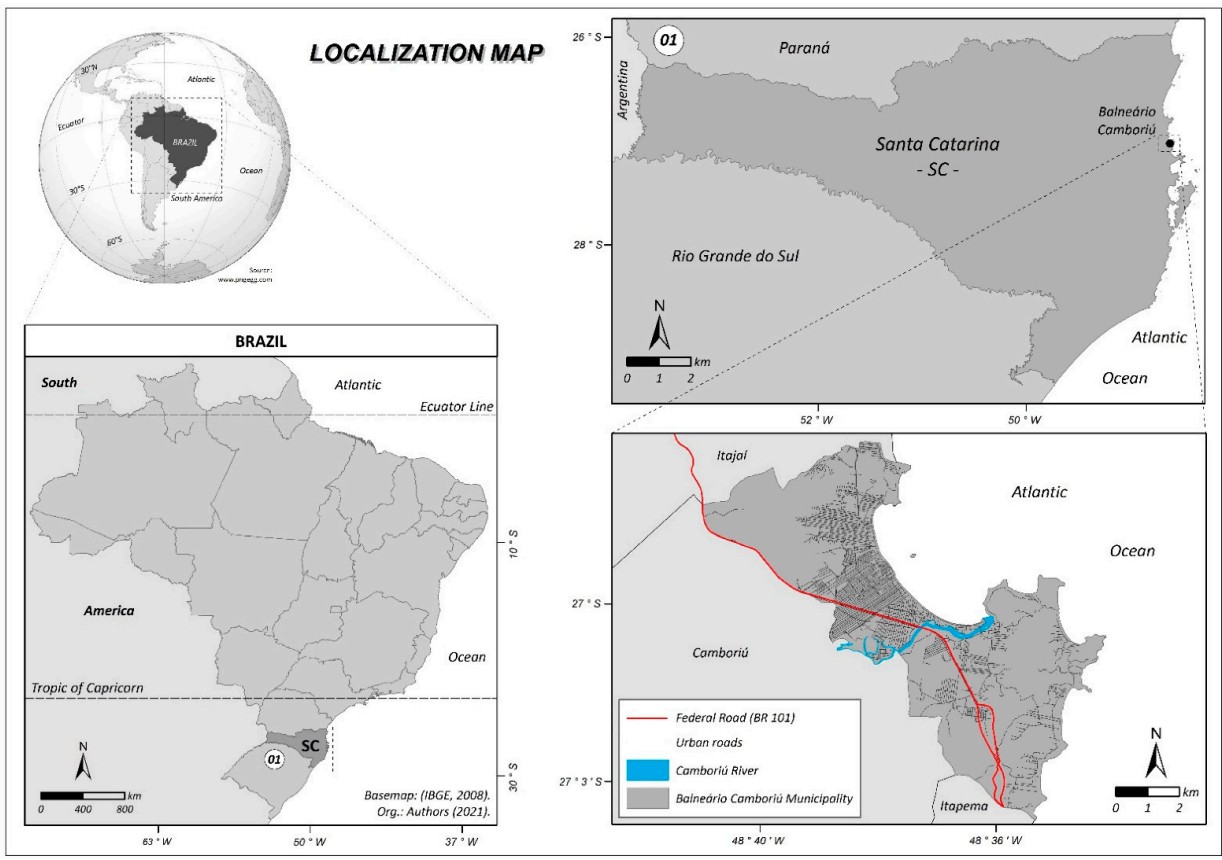

**Figure 1.** Location of the municipality of Balneário Camboriú. Source: [14].

Balneário Camboriú contains the tallest residential towers in Brazil, with buildings over 80 floors high (Figure 2), which earn the city the nickname of "Brazilian Dubai" [18].

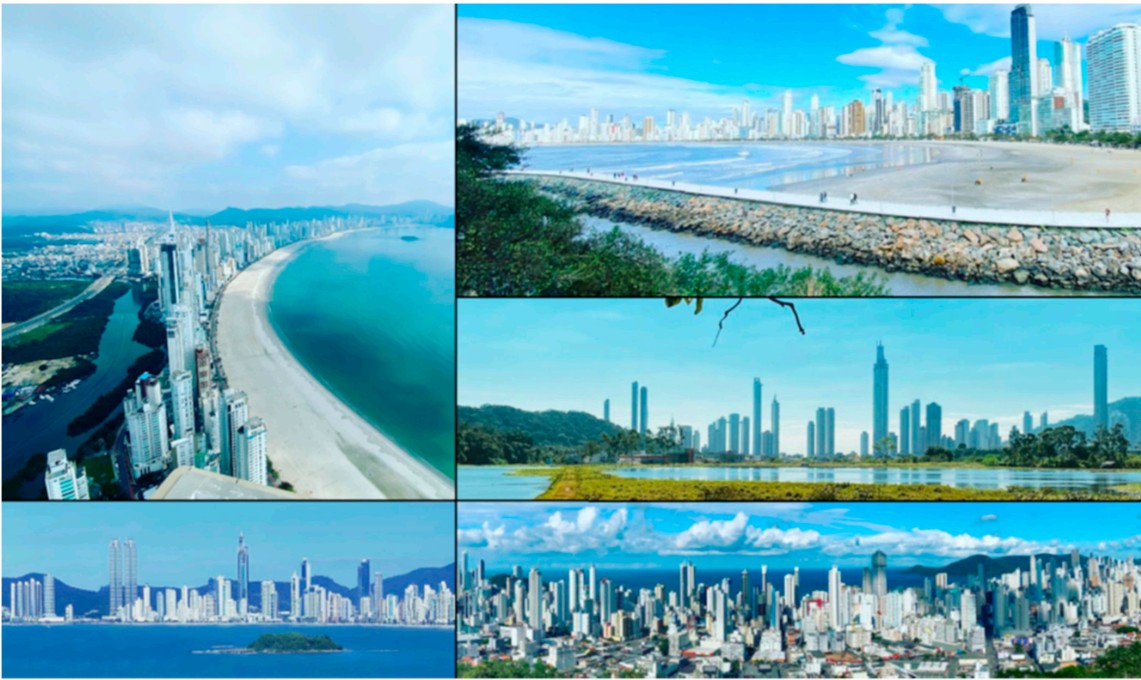

**Figure 2.** Photos of the Balneário Camboriú skyline and its vertical and urbanization pattern. Source: Field works. The authors (2022).

### 2.2. Measuring Devices

Eight fixed sampling points were chosen for this study, seven of them in the urban area and one in the rural area, called Point 0, which corresponds to the official meteorological station of the National Institute of Meteorology (INMET—official code 83898), located in an area surrounded by plantations and mountains, which reflects regional environmental conditions and land use and occupation. This point was selected to serve as a benchmark for calculating the magnitude of heat islands [19,20].

The selected locations represent the most diverse features of the city that influence the urban climate, including urban geometry, land cover, areas with high verticalization and areas with low verticalization, and distance from the city's boundaries [14,21].

Figure 3 displays the features of the seven collecting stations dispersed over the study area. The mini shelters were hung with wires on small poles or structures free of coverings above them, and in areas that represented the urban roughness of Balneário Camboriú. For each of the seven data collection points, in addition to the meteorological station, the name, geographic coordinates, altitude above sea level and a brief description of where the point is located were specified.. In addition, a 100-meter-radius photograph of the spot was examined in order to visualize the ground cover. The images were taken from a viewpoint height of 400 m, according to Google Earth observations.

At each location mentioned in Figure 3, a low-cost, waterproof polypropylene meteorological shelter (LCMS) was installed one and a half meters above ground level respecting the same conditions as the official meteorological station at Point 0. The shelter comprised seven 12-centimetre-diameter plastic plates (used for gardening) and one 18-centimetre-diameter, 2-centimetre-thick spherical white polypropylene plate weighing 30 grams. For durability, the plates were placed between three threaded metal pins, and to keep the same distance between them, a 2-centimetre-long transparent plastic hose was attached between them.

The LCMS was approximately 20 centimetres tall and 18 centimetres in diameter, and it weighed about 0.2 kg. These low-costs meteorological shelters have been tested with positive results [22], and they can be seen in Figure 4.

An HT 500 Instrutherm thermo-hygrometer was utilized to collect data within each LCMS. The HT 500 s were calibrated to gather hourly Ta data from 14 October 2020 to 22 October 2020.

The SVF approach, which consists of photographing at shoulder height with a fisheye lens on a sunny day, was utilized as one of the methods. The photographs were taken in front of the sky [23]. An iPhone 11 Pro Max model with a 12-megapixel camera was utilised to capture these images. As shown in Figure 5, a fisheye lens with 238 apertures was attached to the apparatus.

The SFV was gathered at the same locations that climatic data were gathered. Figure 6 depicts the SFV for every point.

### 2.3. Data Processing and Analysis

In RayMan Pro, the processing and calculation of the SVF of the photos was performed. This program is freely available and is continuously updated and enhanced. It was created for urban climate research with an emphasis on applied climatology [24]. The software's primary simulation inputs are Ta, relative humidity (RH), and wind speed (Ws). In order to estimate the impact of morphological changes, the calculations were based on photos captured with a fisheye lens (Figure 5) and the sun's path. Thus, the method enables examination of the area's many layouts, such as the edge, proximity to green spaces, and urban canyons [24].

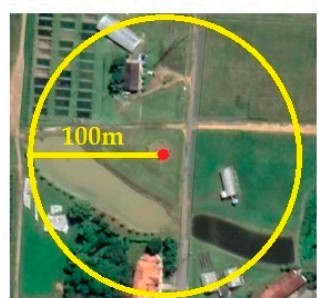

**Point 0**

**Lat:** − 26°57'03.48"
**Long:** − 48°45'43.32"
**Altitude:** 9m

Located in an area, surrounded by plantations and mountains, reflecting the geographical characteristics of the region. Location of INMET/EPAGRI conventional meteorological station for reference of heat islands.

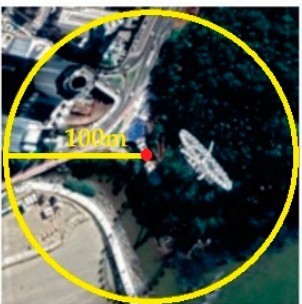

**Point 1**

**Lat:** − 26°58'18.15"
**Long:** − 48°37'54.26"
**Altitude:** 10m

Located on the banks of a stream with exit to the sea, close to the sea, dense vegetation, next to an avenue with intense traffic (people and cars) that gives access to other neighborhoods.

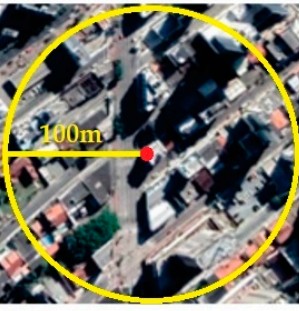

**Point 2**

**Lat:** − 26°59'19.59"
**Long:** − 48°38'21.69"
**Altitude:** 8m

Located next to an avenue with intense traffic (people and cars), surrounded by narrow streets, fully urbanized with buildings (2 to 5 floors). Without the presence of vegetation.

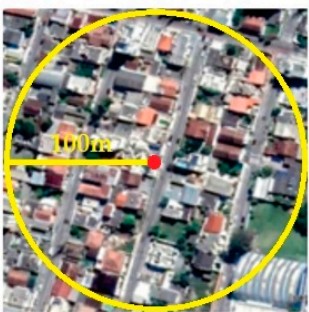

**Point 3**

**Lat:** − 27°00'32.12"
**Long:** − 48°36'38.50"
**Altitude:** 7m

Located in a residential neighborhood, with little shrubbery, with the presence of residences (1 to 2 floors).

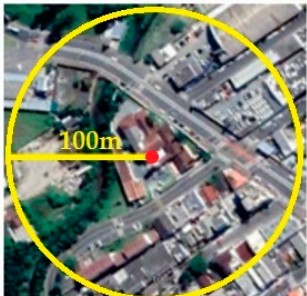

**Point 4**

**Lat:** − 27°00'5.47"
**Long:** − 48°38'41.76"
**Altitude:** 3m

Located on the banks of the Peroba River, surrounded by two avenues, it has high traffic (people and cars), with little shrubbery.

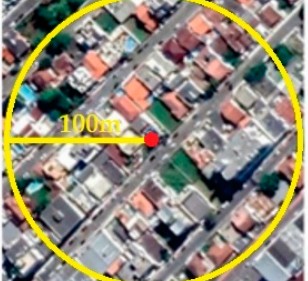

**Point 5**

**Lat:** − 26°59'56.04"
**Long:** − 48°38'9.41"
**Altitude:** 4m

Located in a residential neighborhood, with little shrubbery, with the presence of residences (1 to 2 floors).

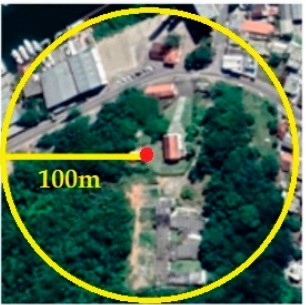

**Point 6**

**Lat:** − 27°00'32.35"
**Long:** − 48°36'16.31"
**Altitude:** 17m

Located near the direct bank of the Camboriú River, dense vegetation, low urban concentration.

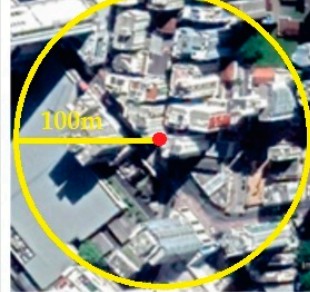

**Point 7**

**Lat:** − 27°00'15.31"
**Long:** − 48°37'16.37"
**Altitude:** 11m

Located next to an avenue with intense traffic (people and cars), surrounded by buildings of (⩾10 floors). Without presence of vegetation.

**Figure 3.** Characteristics of the points selected for the survey. Source: Google Images. Elaborated by the authors (2022).

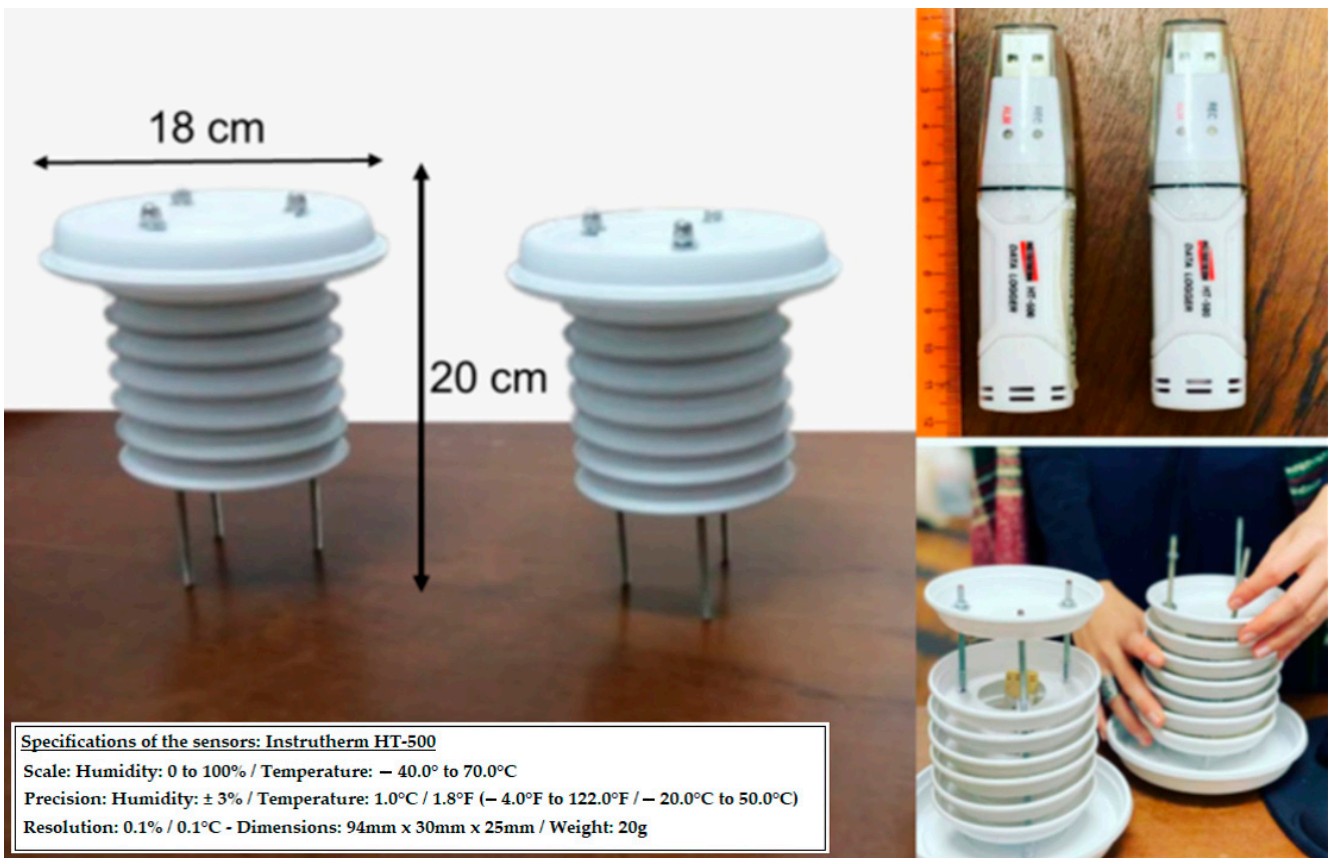

**Figure 4.** LCMS and HT 500 Instrutherm thermo-hygrometer. Source: the authors (2022).

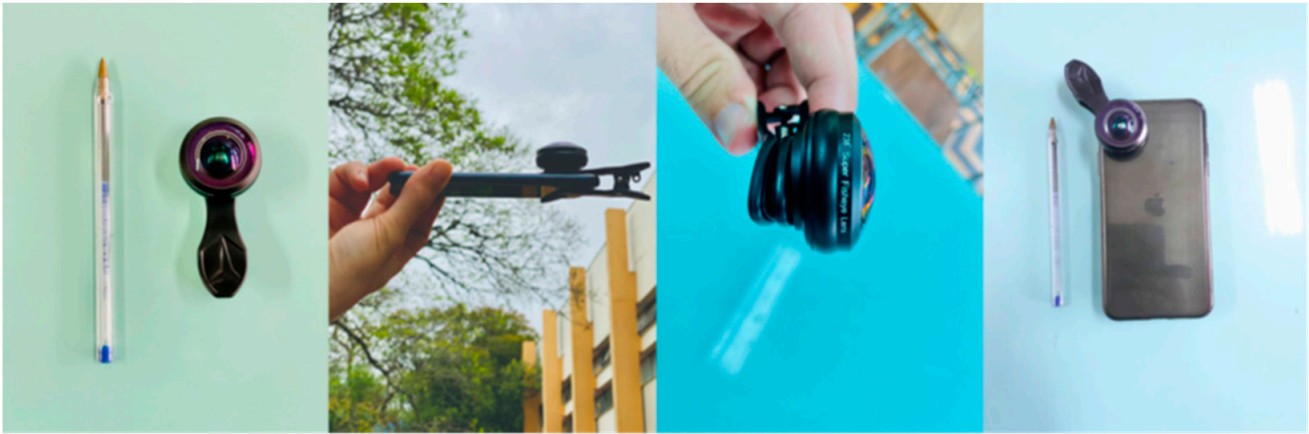

**Figure 5.** Equipment used to perform the SVF. Source: the authors (2022).

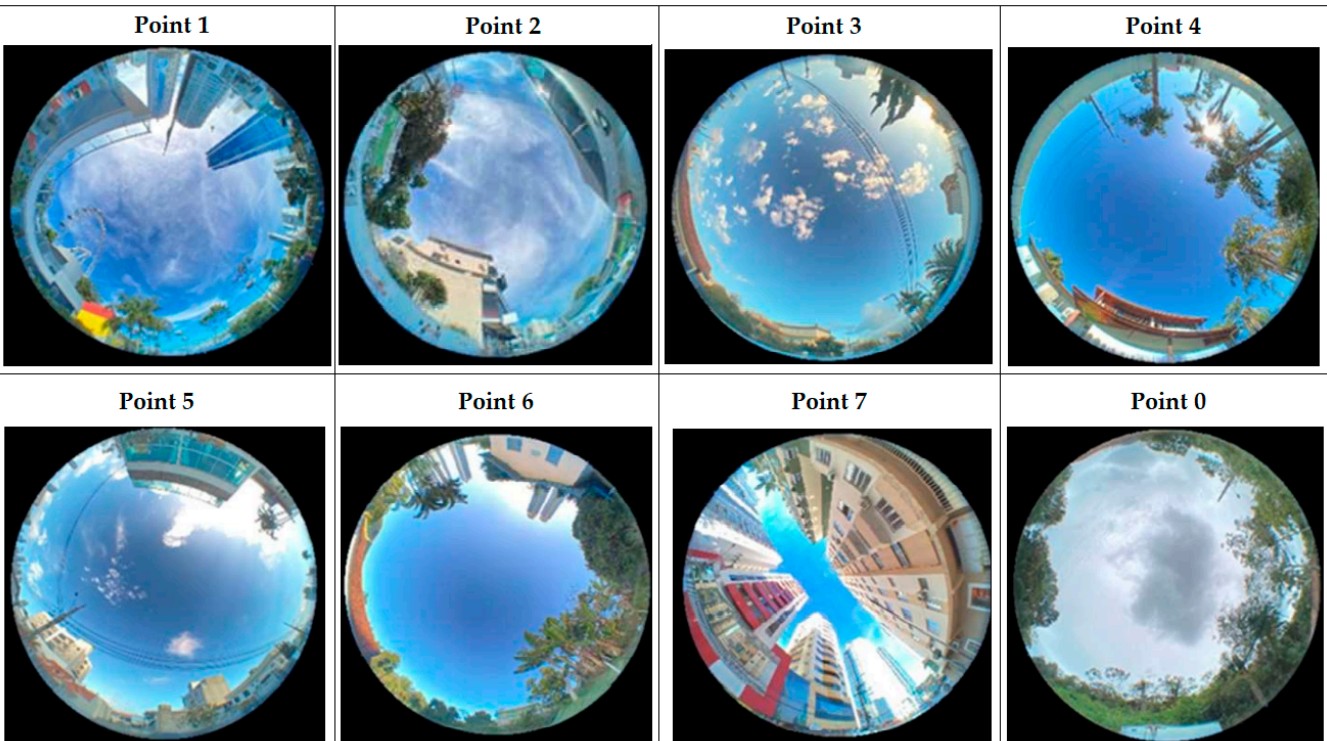

**Figure 6.** SVF of the urban climate data collection points in Balneário Camboriú. Source: The authors (2022).

This research also makes use of the Balneário Camboriú LCZs, which were designed first. This technique has been utilised in a number of global research studies, as well as urban climate studies, particularly UHI studies cities [5,6,25–29]. The method is intended to standardize global urban and rural studies so that they can be compared in the future. In addition to offering evaluation criteria for heat islands, architectural aspects and rural landscapes were utilised.

As indicated in a previous systematic review prepared to further illustrate the methodology of this study, instructions were provided for illustrating the data using the World Urban Database and Access Portal Tools (WUDAPT) approach. This approach downloads the LCZ model file and launches Google Earth [6]. When downloading and opening the file in the specified program, a Landsat 8 image in good condition and devoid of clouds will be chosen. After correcting the image and scale, the following step is to perform the training area, which includes the creation of polygons containing some LCZs. These polygons must include LCZ, and polygon overlap is not permitted [6].

For the final construction of the Balneário Camboriú LCZ map, it was necessary to supply personal information, classification information, and a KML file containing polygons. The WUDAPT Web system performed image quality control, sent confirmation of receipt via e-mail, and, within minutes, sent the map and findings by e-mail and made them accessible to all on the website. Finally, a GIS-based procedure was performed in ArcGIS 10.5, branded by ESRI, to improve the map presentation and scale adjustment for the study area.

In order to study the seven collection points of Balneário Camboriú, the magnitude of the UHI was determined based on the thermal variations between the warmest and coolest points. Differences between the magnitudes of UHI of 2.0 °C were defined as low magnitude, 2.0 °C to 4.0 °C as medium magnitude, and >5.0 °C as higher magnitude [19]. To calculate the UHI intensity [20], the values recorded in the urban area of Balneário Camboriú were obtained and subtracted from the data from the official meteorological station in Itajaí, which is 12 km from the place under study.

## 3. Results

### 3.1. Data Processing and Analysis

The RayMan SVF varies among the collecting points, ranging from 0.126 at point 7 to 0.750 at point 3 and 0.950 in Point 0 [30]. Consequently, the greater the value, the greater the canopy opening, and the lower the value—as depicted in Figure 7—the smaller the canopy opening [10].

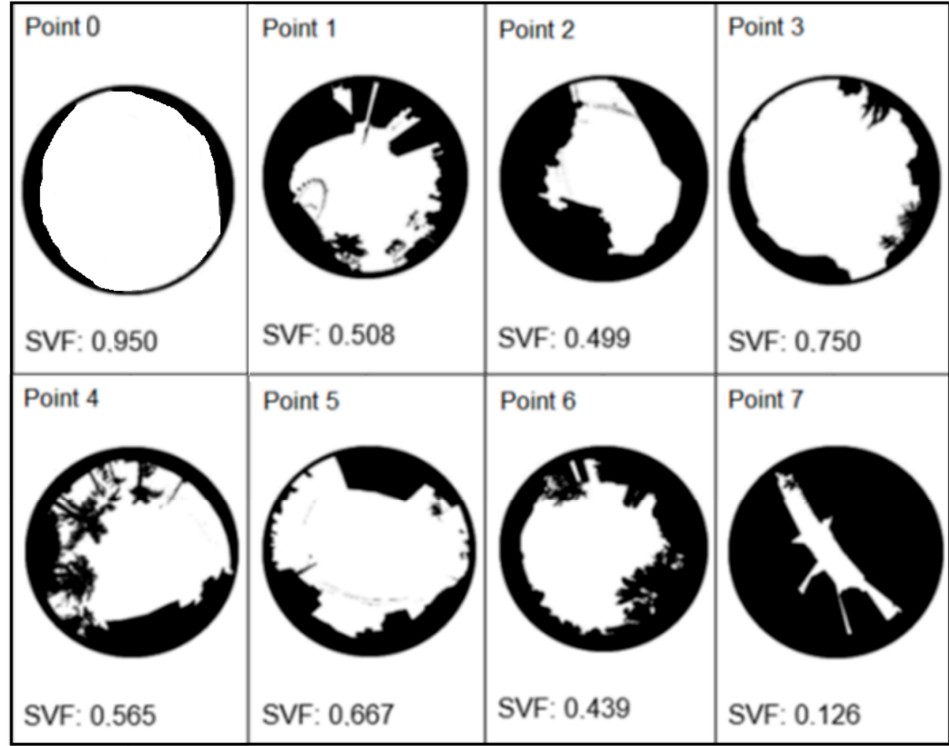

**Figure 7.** RayMan SVF values of collection points. Source: the authors (2022).

The SVF values vary according on urban morphology type (houses, greenery, skyscrapers, family villas, and high-end apartments). At point 01, a variety of urban characteristics, including skyscrapers and vegetation, are visible, and the SVF is 0.508. Point 02 is flanked by mixed residential and commercial structures, with an SVF of 0.499. Point 03 has the greatest SVF (0.750) among the places with limited vegetation and family houses. Point 04 features a configuration of one- to two-story family dwellings and an SVF of 0.663. Point 05 has comparable features to point 03, but with a higher concentration of two- to three-story houses; as a result, its SVF is 0.667, which is slightly lower than that of point 03. It has an SVF of 0.439 and is surrounded by medium to large buildings and surrounding skyscrapers. At point 07, where the SVF is 0.126 and there is a high concentration of buildings, the SVF is the lowest. Point 0, where the weather station is located, is almost totally open.

### 3.2. LCZs of Balneário Camboriú

Using the WUDAPT method, 14 of the 17 types of LCZs were identified in Balneário Camboriú. They are classified as: LCZ 1, LCZ 2, LCZ 3, LCZ 4, LCZ 5, LCZ 6, LCZ 8, LCZ 9, LCZ A, LCZ B, LCZ D, LCZ E, LCZ F, and LCZ G. Its characteristics can be read in Figure 8.

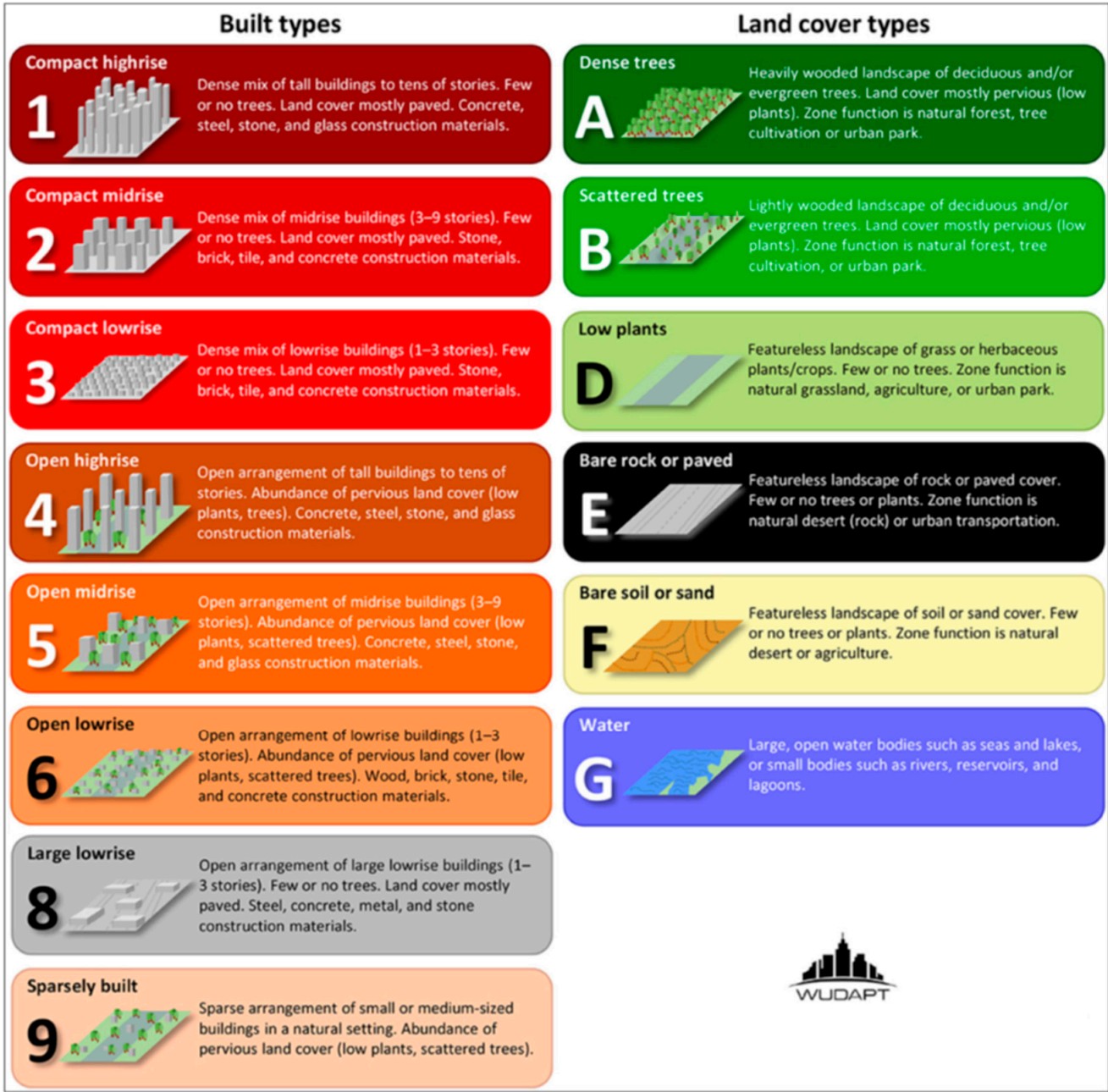

**Figure 8.** Land cover types of LCZs identified in Balneário Camboriú. Source: [6].

In the urban area of Balneário Camboriú, LCZ 1 is close to the Atlantic Ocean and LCZ 03 is further from the Atlantic Ocean. In the surroundings of LCZ 3, it is possible to see the other LCZs—LCZ A, LCZ B and LCZ D—as well as the sand strip of the LCZ F beach and the presence of water represented by LCZ G (Figure 9). Point 0 is of type LCZ D, but it is not included in the map in order to give more prominence to urban points.

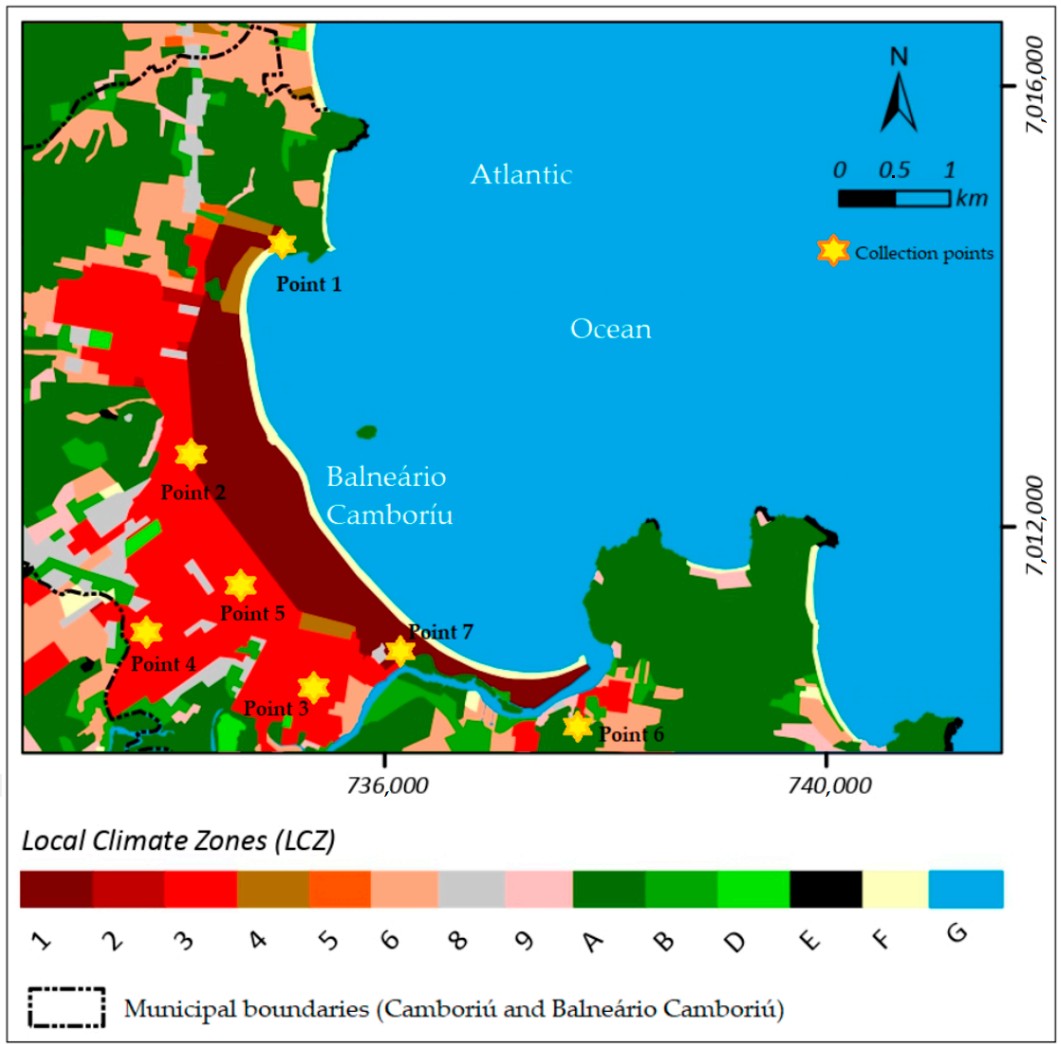

**Figure 9.** LCZ map of Balneário Camboriú and the locations of the points selected for the survey. Source of the cartographic base: City Hall of Balneário Camboriú/SC (2019). Elaboration: the authors (2022).

The Ta collection points for this research are located in the LCZs [5], according to Table 1:

**Table 1.** Ta collection points and LCZs location and description.

| Point | LCZ Class | LCZ Name | Description of LCZ—Land Cover Types |
|---|---|---|---|
| 0 | D | Low plants | Featureless landscape of grass or herbaceous plants/crops. Few or no trees. Zone function is natural grassland, agriculture, or urban park. |
| 1 | 4 | Open highrise | Open arrangement of tall buildings with dozens of floors. Plenty of permeable ground cover (low plants, scattered trees). Concrete, steel, stone, and glass building materials |
| 2 and 7 | 1 | Compact highrise | Dense mix of tall buildings with dozens of floors. Few or no trees. Ground cover mostly paved. Concrete, steel, stone, and glass building materials |
| 3, 4 and 5 | 3 | Compact lowrise | Dense mix of low-rise buildings (1 to 3 floors). Few or no trees. Ground cover mostly paved. Stone, brick, tile, and concrete building materials. |
| 6 | B | Scattered trees | Slightly wooded landscape of deciduous and/or evergreen trees. Mainly permeable ground cover (low plants). The function of the zone is natural forest, tree cultivation or urban park |

Source: The authors (2022).

### 3.3. Spatial Variation, Thermal Range and UHI

The nine days of analysis were divided into groups of three days in order to better facilitate the visualization of the data; the graphs in Figure 10 refer to the hourly collections of the seven points between 14 October and 16 October 2020.

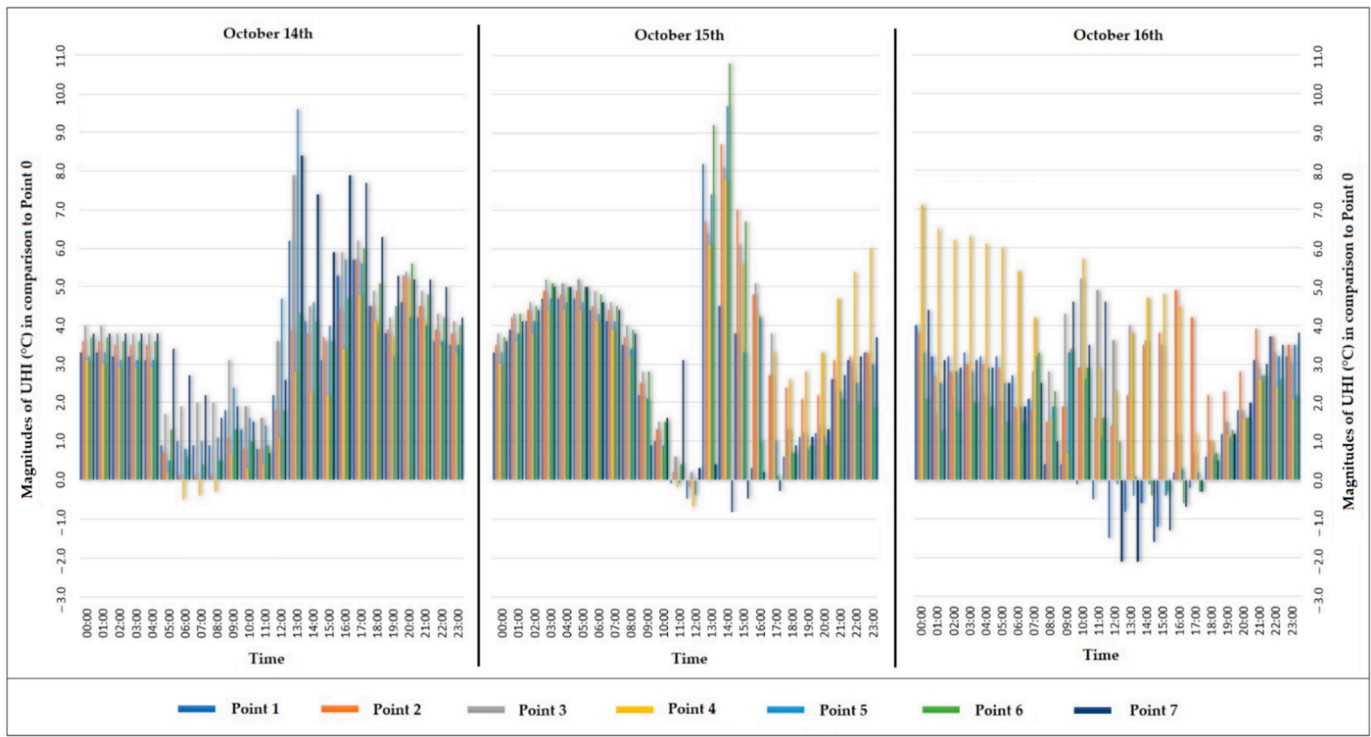

**Figure 10.** Hourly magnitude of the Ta and UHI of the seven points in Balneário Camboriú between 14 October and 16 October 2020. Source: the authors (2022).

On 14 October 2020, during the period between 00:00 and 04:00, there was no significant variation in the magnitudes of the UHI between the points, but throughout the time there was a UHI with a magnitude of approximately 4.0 °C; at this time of day, the highest magnitudes of the UHI were observed at points 3 and 7.

During the morning (05:00 to 12:00), there was a reduction in the magnitude of the UHI at several locations, most notably at point 4, which displayed a magnitude of near −1.0 °C. From 13:00 to 17:00, the urban heat islands grew in comparison to the early morning, reaching 9.0 °C at point 5. Point 7 also displayed a UHI of considerable magnitude relative to the other sites. Point 07 has the maximum magnitude of UHI at night after 18:00, reaching 5.0 °C at 22:00.

On 15 October 2020, between 00:00 and 04:00, the amplitude between urban and rural areas grows everywhere. Therefore, from 05:00 to 12:00, the UHI intensities behaved similarly to the previous day, with a decrease. Points 01, 04, 02, and 05 indicate islands of relative coolness with a magnitude of around −1.0 °C [19,20]. The heat islands at these points expanded between 13:00 and 17:00, with the exception of point 07, which lowered and registered an island of coolness. With the exception of points 04 and 07, the magnitude of the UHI decreased from 18:00 to 23:00, with point 04 exhibiting a UHI magnitude of 6.0 °C.

Point 04 exhibited maximum amplitudes on 16 October 2020, from 00:00 to 05:00, with amplitudes larger than 7.0 °C. From 06:00 to 12:00, the amplitudes between this interval and the other points varied significantly. In other words, point 4 exhibited the greatest thermal amplitudes. Point 7 had negative amplitudes at this time. Between 1:00 and 5:00 on the same day, points 1, 5, 6, and 7 had cool islands of magnitude more than −2.0 °C, whereas points 2, 3, and 4 had cool islands of magnitude 4.0 °C, due to the shading effect

of the built environment. At night (after 18:00), the points with coolness islands began to record heat islands with a lesser magnitude than the other sites.

Figure 11 refers to the hourly collections at the seven points between 17 October and 19 October 2020.

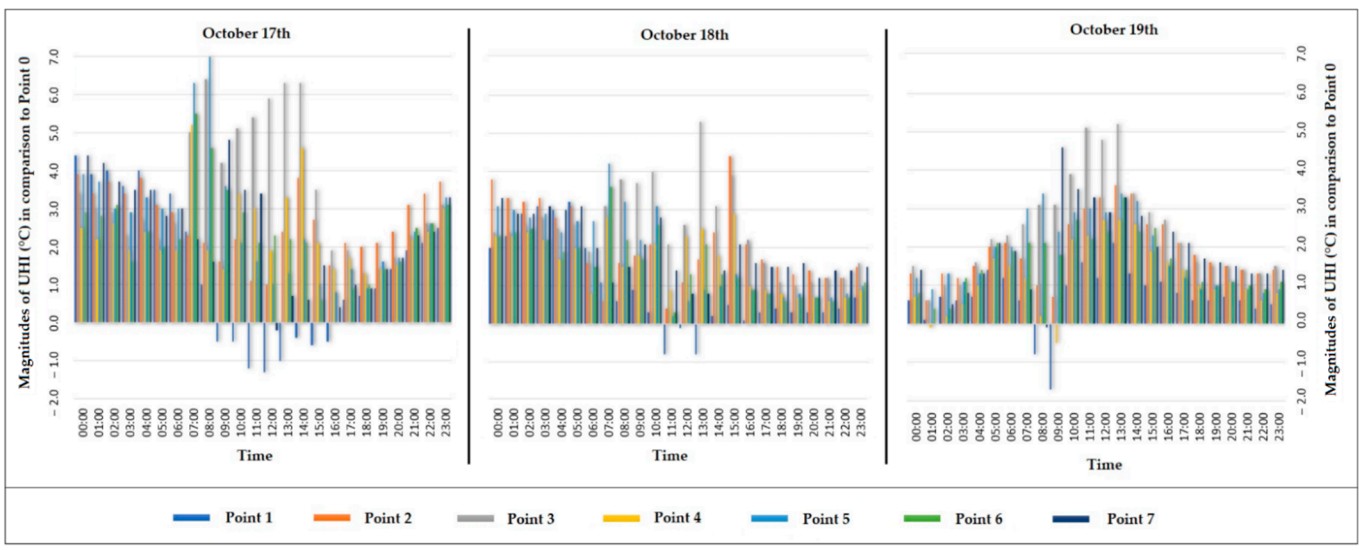

**Figure 11.** Hourly magnitude of the Ta and UHI of the seven points in Balneário Camboriú, between 17 October and 19 October 2020. Source: The authors (2022).

During the period from 00:00 to 05:00 on 17 October 2020, there was little variation in the magnitudes of the heat islands between the points. However, from 06:00 to 12:00, there was a large variation in the magnitudes of the heat islands between the points, with point 5 presenting a UHI with a magnitude of 7.0 °C and point 01 at 10:00 presenting a coolness island with a magnitude greater than −1.0 °C.

During the time from 1:00 p.m. to 5:00 p.m., the magnitude of the UHI varied greatly between the sites, with point 3 exhibiting the maximum magnitude of the heat islands, reaching 6.0 °C, and point 1 exhibiting islands of coolness of roughly −1.0 °C. At point 3, the magnitude of the UHI falls after 18:00, whereas at the other places, the magnitude of the UHI grows in respect to the afternoon.

On 18 October 2020, from 00:00 to 06:00, there was little fluctuation in the amplitude of the UHI between the points, comparable to the trend observed on prior days. In the early morning (after 6:00), the differences between the sites were noteworthy, especially point 01, which has a cool island of nearly −1.0 °C, and points 3 and 5, which had the greatest UHI magnitudes, exceeding 5.0 °C.

Between 12:00 and 17:00, the points continued to exhibit significant variations, especially points 2 and 3, which had the highest magnitudes (approximately 5.0 °C), and point 1, which presented the lowest magnitude, although during this period it began to register positive magnitudes, no longer an island of coolness as it had been during the preceding period. At night (after 18:00), there was not a huge magnitude difference between the points, and their heat islands were less than 2.0 °C in magnitude.

On 19 October 2020, from 00:00 to 06:00, the UHI magnitudes remained modest (not reaching 2.0 °C) and there was little fluctuation between the points. Point 04 registered a freshness island with a magnitude of less than −1.0 °C throughout the same period. The magnitudes of the points experienced considerable variations as a result of the presence of solar radiation at 06:00, and there were points with islands of freshness (points 1, 4, and 7). Points 3 and 7 had the greatest heat islands, with magnitudes of around 5.0 °C. Point 7 occasionally featured regions of coolness and islands of heat. After 12:00, the magnitudes of

the heat islands diminished, and there was no significant variance across the spots. As with the previous day, the magnitudes of the UHI decreased after 18:00, not exceeding 2.0 °C.

Figure 12 refers to the hourly collections of data from the seven points between 20 October and 22 October 2020.

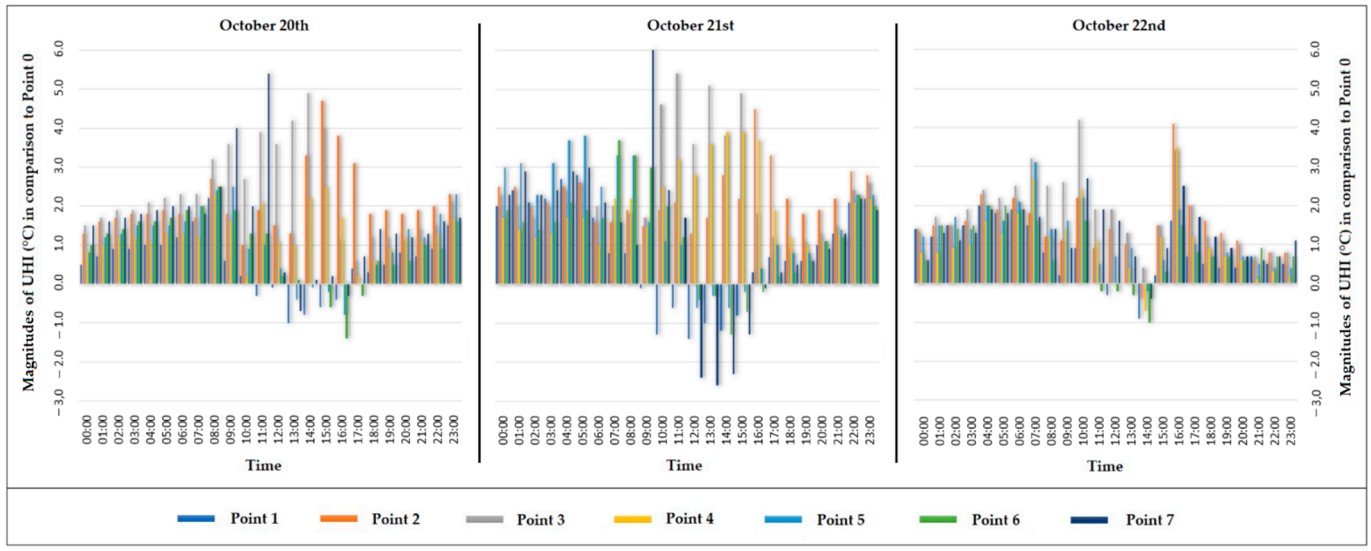

**Figure 12.** Hourly magnitude of the Ta and UHI of the seven points in Balneário Camboriú between 20 October and 22 October 2020. Source: the authors (2022).

Beginning at midnight on 20 October 2020, the graph has the same features as the previous day. Between the hours of 06:00 and 17:00, variations in the heat islands' magnitude were presented. In this timeframe, the magnitudes of the UHI were pronounced, mainly in the case of point 7 (11:00 AM on October 20th, 2020), which showed the greatest variation in this period, in addition to having the second highest UHI (5.0 °C) during the night and, in a few hours, an UFI (−1.0 °C).In the afternoon, the temperatures at points 1, 5, 6, and 7 were below −1.0 °C. After 18:00, the magnitudes did not exhibit substantial changes, and there were no islands of coolness, only heat islands with magnitudes between 1.0 °C and 2.0 °C.

On 21 October from 06:00 to 12:00, there was greater variation between the points, and again point 07 presented the highest magnitudes of UHI (6.0 °C), while point 01 presented a freshness island with a magnitude of −1.0 °C. In the afternoon, between 13:00 and 17:00, points with UHI and urban cool/fresh islands (UFI) were again presented, particularly at points 1, 5, 6, and 7, with the lowest magnitudes of UFI exceeding −2.0 °C, while the highest magnitudes (UHI) were recorded at points 02, 03, and 04, exceeding a magnitude of 5.0 °C. After 18:00 PM, the lowest amplitudes of the day were recorded..

Finally, on 22 October during the period from 07:00 to 12:00, the magnitudes of the UHI presented great variation, particularly point 3, which registered a UHI of magnitude 4.0 °C; point 01 registered a UFI of −1.0 °C. Between 13:00 and 17:00, UHI and cool islands were recorded, especially at point 2, which in the early afternoon presented a cool island (−1.0 °C), while at 16:00 this same point registered the largest UHI of the period under analysis, exceeding a magnitude of 4.0 °C. During the period from 18:00 to 23:00, the magnitudes between the points did not register great variations.

## 4. Discussion

### Impact of Urban Morphology

During the examination of the data gathered in Balneário Camboriú, SC, it was observed that thermal conditions between 00:00 and 06:00 displayed the occurrence of heat islands with different magnitudes. According to urban morphology, the SVF, and the LCZ in which the point is placed, point 0 is in LCZ D; point 01 is located in an area of LCZ 4;

point 2 is in LCZ 1; point 03, point 04 and point 05 are located in an area of LCZ 3; point 6 is in LCZ B; and point 7 is also in LCZ 1. During the cooling period (night), the spots with the smallest SVF present higher temperature data, since these locations hold temperatures better [31].

On October 14th, 2020, which provided a −1.0 °C UFI caused by the proximity of point 04 to a body of water [32,33], the increased availability of water was responsible for the evaporation and retention of latent heat, which provided cooling. In general, between 06:00 and 12:00, the magnitudes of the UHI did not follow a pattern, as there were numerous days with both high- and low-magnitude points. These collection points had a bigger growing amplitude than the other collection places. The more pronounced heating was justified by the location, as the points are in LCZ 3, a warmer place caused by its own urban characteristics only [26,34–38], because, from a geomorphological and altimetric point of view, all of these points are located at altitudes close to the sea, and altimetry is not a cause of this spatial and temporal variation of Ta.

Points 01, 04, 05, 06, and 07 exhibited pockets of freshness at various times during the analysis period. Due to the shading of skyscrapers, the existence of bodies of water, and other factors, urban regions must have cooler Ta than rural areas [11,14,39].

Point 07 has the greatest positive magnitude for this period of analysis as a result of its location within an urban canyon. The larger the urban canyon, the lower the SVF value, as the dissipation of long-wave radiation and the reduction of Ws result in higher Ta values [40]. Referring to the same time period in the morning, it can be seen that the amplitude tends to decrease by the factor of the shaded collecting point.

The shading effects created by tall structures result in a delay in the natural nocturnal cooling process, and high urban temperature values are recorded at times when they are unusual, due to the daytime trapping of longwave radiation in urban canyons, whose energy surplus will be released hours after the onset of negative radiation balance [13,14,41]. Between 13:00 to 17:00 h, the behaviour of the magnitudes of the Ta, also subject to considerable fluctuations due to the geomorphology of the metropolitan area, were recorded around 11.0 °C UHI at point 06. These high magnitudes of UHI at the collection points were closely related to the urban morphology, the SVF, and the LCZ where the individual points were located. These were LCZs 1, 3, densely populated areas with a variety of buildings and no or few trees. Their terrain was predominantly asphalt. Stone, brick, tile, and concrete construction supplies were used in these areas [5].

This is also due to the low SVF value, which is the lowest among all collection sites. Among the consequences of a low SVF in an urban environment is the trapping of heat absorbed by urban structures throughout the day in these locations, as indicated by large-scale studies [10,14,31,42], even if the heat was transported from other points of the city, due to the closure by the urban canyon, the heat can be trapped in these places, increasing the values and magnitudes of the UHI. During this time period, moderate UFI were discovered [19,20,29]. Due to the shading of skyscrapers, the existence of water bodies, and other factors, it was anticipated that air temperatures in dense urban regions would be lower than in rural areas [11,14,39]. During the period from 18:00 to 23:00, there were no UHI with magnitudes of more than 3.0 °C. The LCZ present at the point and the SVF cause these locations to retain greater heat at night [40]. The lower the SVF number, the less dispersion of wave radiation over long distances, which, along with decreasing wind speeds, causes Ta to rise. LCZ 1 and 3, result in the highest Ta values caused by urban elements [34–36].

## 5. Conclusions

In this study the hourly Ta data were recorded at seven points in Balneário Camboriú urban areas. As Balneário Camboriú grows vertically, there was a large variation in the SVF of the collection points, with some points having an SVF close to 0 and others having an opening greater than 0.650, which is a key factor in the generation of heat islands and the formation of freshness islands in the studied area. The LCZ were included for the collection points and also contributed to the formation and duration of the heat islands. The collection points with the highest UHI magnitudes were located in LCZ 1 and LCZ 3, and the points with the lowest magnitudes were recorded in LCZ B and LCZ 4.

Considering the findings of the data analysis on the magnitude of the heat islands in Balneário Camboriú, it was clear that the magnitude varies according to the urban morphology, the SVF, and the LCZ. Moreover, in this investigation, a UHI with a maximum magnitude of 10.8 °C and fresh islands with a magnitude of −4.5 °C were identified.

This is a preliminary investigation, a first attempt. The main goal of the research group in this study was not to make new findings about urban climate, but rather to present a small portion of the Ta fluctuation, LCZ, and SVF in a highly verticalized metropolis that had never been researched before. Additional in-depth study has been conducted, and we intend to publish more in-depth studies with news about this unique Brazilian urban setting in the future.

**Author Contributions:** Conceptualization, C.A.W. and I.L.H.; methodology, C.A.W., I.L.H., J.P.A.G. and S.S.; software, C.A.W. and A.S.B.; validation, I.L.H., C.A.W., J.P.A.G., S.S. and A.S.B.; formal analysis, J.P.A.G. and S.S.; investigation, I.L.H. and C.A.W.; resources, C.A.W. and J.P.A.G.; data curation, C.A.W., J.P.A.G. and A.S.B.; writing-original draft preparation, C.A.W. and I.L.H.; writing—review and editing, C.A.W., I.L.H., J.P.A.G. and S.S.; visualization, C.A.W., J.P.A.G. and S.S.; supervision, C.A.W., J.P.A.G. and S.S.; project administration, C.A.W.; funding acquisition, C.A.W. All authors have read and agreed to the published version of the manuscript.

**Funding:** This study was financed in part by the Coordenação de Aperfeiçoamento de Pessoal de Nível Superior, Brasil (CAPES), Finance Code 001.

**Informed Consent Statement:** Not applicable.

**Acknowledgments:** We thank the Conselho Nacional de Desenvolvimento Científico e Tecnológico (CNPq) for providing the Research and Productivity research grant process number 306505/2020-7.

**Conflicts of Interest:** The authors declare no conflict of interest. The authors declare that they have no known competing financial interests or personal relationships that could have appeared to influence the work reported in this paper.

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
