# Peer review of "Local Climate Zones, Sky View Factor and Magnitude of Daytime/Nighttime Urban Heat Islands in Balneário Camboriú, SC, Brazil"

_climate, doi:10.3390/cli10120197_

Round 1
Reviewer 1 Report
The language and sentence structure are not acceptable even for an early draft (starting with the very first sentence of the abstract). This manuscript should have been proof-read before submission.
The paper is mainly a compilation and basic analysis of measured data. The scientific content is insufficient. The innovative content is low.
The detailed description of the sensors should go to annex.
A single fisheye image at each measurement point does not properly represent the corresponding LCZ. You need to use GIS data and numerically estimate an average SVF for an extended area.
In the section on Data Processing and Analysis, you do not mention at all what Rayman Pro actually does. There is no mention anywhere in the manuscript of mean radiant temperature.
Figure 8 may be problematic due to copyright issues.
Author Response
Dear reviewer,
Please, find attached the answers for your comments.
Thanks for your support.
Best regards.
Authors.

Reviewer 2 Report
Dear Authors,
Please refer to the review report.
Best,

Author Response

(The authors gave the same response as above.)

Reviewer 3 Report
This study determined Local climate zones, sky view factor and magnitude of daytime/nighttime urban heat islands in Balneário Camboriú, SC, Brazil. Relevant topics have been conducted worldwide. The conventional method and bland results led to the novelty of this study was not sufficient in the present form. After carefully evaluating the manuscript, there are some concerns and issues that are related to the clarity of the paper:
1. The authors should rearrange the abstract, the present form is unreadily, the intention, methodology, and main results should be summarized.
2. A comprehensive literature review should be added to clearly reflect: 1) what the relevant research progress is and, 2) why your proposal is important.
3. Novelty unclear: What is the original contribution of the study? The introduction section is not very enlightening on the subject.
4. How the instruments were fixed? What is the surrounding environment, i.e, landscape elements combinations? All these should be addressed in detail.
5. The discussion part should be rearranged, and limitation of the study should be highlighted.
6. Most of the results are common knowledge, and have no means to enhance the present knowledge base.
Author Response
Dear reviwer. We appreciate the suggestions. Thank you so much. Answers are attached.

Round 2
Reviewer 2 Report
Dear Authors,
Thank you for your point-by-point answers.
I realize that my suggestions have been fully realized and I am pleased, above all, for the possibility of dialogue.
Great that you emphasized that geomorphology is not a notable climate control for the coastal city.
I am pleased that they know how important wind direction and speed are at the climate scale in question, thus validating my concern. May these issues be elucidated in subsequent studies.
I regret that the journal does not allow display on horizontal boards, but making the supplementary files available will greatly enhance your results. As a suggestion for future works, in the form of presentation of results: temporal and spatial panels and surface graphs may be options. That is the suggestion.
Minor corrections, please check:
-> Line 263: Table 1, insert its authorship and date.
-> Line 404: insert the period (dot).
-> Finally, the authors [19, 20, 29] use the concept of the magnitude proposed by Fernández García (1996) who treats the intensity of the UHI as a quantitative concept and the magnitude as qualitative.
Congrats!
Author Response

(The authors gave the same response as above.)

Reviewer 3 Report
The authors have made considerable revisions according to the reviewers' comments; this revised version has been substantially improved and therefore can be accepted.